# Survival Improvement over Time of 960 s-AML Patients Included in 13 EORTC-GIMEMA-HOVON Trials

**DOI:** 10.3390/cancers12113334

**Published:** 2020-11-11

**Authors:** Safaa M. Ramadan, Stefan Suciu, Marian J. P. L. Stevens-Kroef, Roelof Willemze, Sergio Amadori, Theo de Witte, Bob Löwenberg, Petra Muus, Boris Labar, Liv Meert, Gaetan de Schaetzen, Giovanna Meloni, Giuseppe Leone, Marco Vignetti, Jean-Pierre Marie, Michael Lübbert, Frédéric Baron

**Affiliations:** 1EORTC Headquarter, 1200 Brussels, Belgium; safaa.ramadan@ieo.it (S.M.R.); stefan.suciu@eortc.org (S.S.); liv.meert@eortc.org (L.M.); gaetan.deschaetzen@eortc.be (G.d.S.); 2Currently in the Hemato-oncology Division, European Institute of Oncology, 20141 Milan, Italy; 3Currently in the Department of Medical Oncology, NCI-Cairo University, 11796 Cairo, Egypt; 4Department of Human Genetics, Radboud University Medical Centre, 6525 GA Nijmegen, The Netherlands; Marian.Stevens-Kroef@radboudumc.nl; 5Hematology, Leiden University Medical Center, 2300 RC Leiden, The Netherlands; r.willemze@outlook.com; 6Hematology, University Tor Vergata, 00133 Rome, Italy; sergio.amadori1946@gmail.com; 7Department of Hematology, Radboud University Medical Centre, 6525 GA Nijmegen, The Netherlands; Theo.deWitte@radboudumc.nl (T.d.W.); p.muus@kpnplanet.nl (P.M.); 8Hematology, Erasmus University Medical Center Rotterdam, 3000CA Rotterdam, The Netherlands; b.lowenberg@erasmusmc.nl; 9Department of Hematology, Kings College Hospital, London SE5 9RS, UK; 10Department of Hematology, University Hospital Center Rebro, 10000 Zagreb, Croatia; boris.labar@inet.hr; 11Department of Cellular Biotechnologies and Hematology, Sapienza University, 00161 Rome, Italy; meloni@bce.uniroma1.it; 12Istituto di Ematologia, Università Cattolica S. Cuore, 00168 Rome, Italy; giuseppe.leone@unicatt.it; 13Data Center and Health Outcomes Research Unit, Italian Group for Adult Hematologic Diseases (GIMEMA), 00182 Rome, Italy; m.vignetti@gimema.it; 14Department Hematology and Tumor Bank, Saint-Antoine Hospital, AP-HP & University Pierre & Marie Curie, 75571 Paris, France; mariejeanp@gmail.com; 15Department of Hematology, Oncology and Stem Cell Transplantation, Freiburg University Medical Center, Faculty of Medicine, 79106 Freiburg, Germany; michael.luebbert@uniklinik-freiburg.de; 16GIGA-I3, Hematology, University of Liège, 4000 Liège, Belgium; 17Hematology, CHU of Liège, 4000 Liège, Belgium

**Keywords:** AML, secondary, trials

## Abstract

**Simple Summary:**

Secondary acute myeloid leukemia (s-AML) refers to the development of AML following myelodysplatic syndrome or other hematological malignancies, or after a solid tumors, or nonmalignant diseases or following exposure to environmental or occupational carcinogens. Here, we report data from 960 s-AML patients who were treated in 13 EORTC collaborative trials conducted between May 1986 and January 2008. The main aims of our study were (1) to assess whether overall survival of s-AML patients improved over time, (2) to identify initial disease features associated with overall survival. We observed that overall survival of younger patients improved over the years, in parallel with introduction of high-dose cytarabine in induction remission chemotherapy. This suggests that this strategy should be further investigated in younger patients with s-AML. Furthermore, this study confirmed that the sAML patients having adverse cytogenetic risk features and those with high white blood cells at diagnosis had a dismal survival, regardless of their age group.

**Abstract:**

We report the outcomes of secondary acute myeloid leukemia (s-AML) patients included in one of 13 European Organisation for Research and Treatment of Cancer (EORTC) collaborative AML trials using intensive remission-induction chemotherapy. Among 8858 patients treated between May 1986 and January 2008, 960 were identified as having s-AML, either after MDS (cohort A; *n* = 508), occurring after primary solid tumors or hematologic malignancies other than MDS (cohort B; *n* = 361), or after non-malignant conditions or with a history of toxic exposure (cohort C; *n* = 91). Median age was 64 years, 60 years and 61 years in cohort A, B and C, respectively. Among patients ≤60 years and classified in the cohorts A or B (*n* = 367), the 5-year overall survival (OS) rate was 28%. There was a systematic improvement in the 5-year OS rate over three time periods (*p* < 0.001): 7.7% (95% CI: 1.3–21.7%) for patients treated before 1990 (period 1: *n* = 26), 23.3% (95% CI: 17.1–30.0%) for those treated between 1990 and 2000 (period 2: *n* = 188) and 36.5% (95% CI: 28.7–44.3%) for those treated in 2000 or later (period 3: n = 153). In multivariate analysis, male gender (HR = 1.39; *p* = 0.01), WBC ≥ 25 × 10^9^/L (HR = 2.00; *p* < 0.0001), age 46-60 years (HR = 1.65; *p* < 0.001) and poor-risk cytogenetics (HR = 2.17; *p* < 0.0001) were independently associated with shorter OS, while being treated during period 2 (HR = 0.50, *p* = 0.003) or period 3 (HR = 0.43; *p* = 0.0008). Having received high-dose cytarabine (HD-AraC) (*n* = 48) in the induction chemotherapy (HR = 0.54, *p* = 0.012) was associated with a longer OS. In contrast, among patients >60 years of age (*n* = 502), the OS was dismal, and there was no improvement over time.

## 1. Introduction

Secondary acute myeloid leukemia (s-AML) refers to the development of AML following a history of malignant or nonmalignant diseases (such as autoimmune diseases) or following exposure to environmental or occupational carcinogens [1,2]. Population-based studies have revealed that s-AML could represent up to 25% of AML [3]. This rate will most likely further increase with the improvement in cancer survival rates. S-AML can be separated in three distinct groups: (1) AML occurring after myelodysplatic syndrome (MDS), (2) AML occurring after other hematological malignancies or solid tumors and (3) AML occurring after nonmalignant disease and or toxic exposure.

S-AML has been associated with poorer outcomes than de novo AML. Potential reasons include frequent clonal hematopoiesis and depletion of hematopoietic reserve leading to slower hematologic recovery following intensive remission-induction chemotherapy, the possibility of primary disease recurrence in case of s-AML occurring after a prior malignant disease or MDS and higher incidence of poor-risk cytogenetics and of TP53 mutations in s-AML than in de novo AML patients [4,5,6].

Knowledge on s-AML is mostly derived from retrospective studies with relatively low numbers of patients and from National Cancer Registries [7]. Here, we report data from an EORTC database, which includes data from s-AML patients who were treated in 13 EORTC collaborative trials conducted between May 1986 and January 2008. The main aims of our study included (1) to assess whether outcome of s-AML patients improved over time, (2) to identify prognosis factors of s-AML and (3) to assess the impact of induction intensity and of allogeneic hematopoietic stem cell transplantation (allo-SCT) on the outcomes of s-AML patients.

## 2. Results

### 2.1. Patients and Overall Survival (OS) in the Whole Cohort

A total of 8858 AML patients were treated in one the 13 selected EORTC collaborative AML trials (Appendix A). All patients were treated with intensive remission-induction chemotherapy. Among them, 960 (11%) were identified as having s-AML. This included 508 patients (6%) who had s-AML after MDS (cohort A), 361 patients (4%) who had s-AML occurring after primary solid tumors or hematologic malignancies other than MDS (cohort B) and 91 patients (1%) in whom s-AML developed after non-malignant conditions, such as autoimmune disease or following a history of toxic exposure (cohort C) (Appendix A). Median ages were 64 years, 59 years and 61 years in cohorts A, B and C, respectively (*p* < 0.001). A total of 412 patients were ≤60 years at the start of AML treatment, while the remaining 548 patients were 61–85 years old. Median follow-up was 5.8 years in patients ≤60 years, and 5.4 years in those >60 years.

The 5-year overall survival (OS) rate was significantly lower in patients from cohort A (10.6%, 95% CI: 7.9–13.8%) than in those from cohort B (22.1%, 95% CI: 17.8–26.8%) or C (22.5%, 95% CI: 14.4–31.7%) (*p* = 0.062; Appendix A). As expected, the 5-year OS rate was significantly higher in patients ≤ 60 (27.9%, 95% CI: 23.4–32.5%) than in those >60 years (7.4%, 95% CI: 5.2–10.0%) (*p* < 0.0001; Appendix A). Furthermore, interestingly, the 5-year OS rate improved significantly over time: from 8.5% (95% CI: 4.2–14.8%) for patients included <1990 to 20.5% (95% CI: 16.3–25.0%) in patients included ≥2000 (*p* < 0.0001; Appendix A). Since only 91 patients were included in cohort C, we elected to focus our analyses on 869 patients from cohorts A and B. There were 367 patients ≤60 years old and 502 patients older than 60 years.

### 2.2. Outcomes in Patients ≤ 60 Years Old with Secondary Acute Myeloid Leukemia (s-AML) Occurring after MDS or Other Malignancies

#### 2.2.1. General Characteristics and Outcomes

The baseline characteristics of younger patients with s-AML following MDS (cohort A, *n* = 181) or following other malignancies (cohort B, *n* = 186) are summarized in Table 1. Among other malignancies, breast (*n* = 55), genitourinary (*n* = 17), gynecologic (*n* = 16), head and neck (*n* = 15) and skin (*n* = 10) cancers were the most frequent solid tumors (*n* = 131), while Hodgkin’s lymphoma (*n* = 26) and mature B-cell neoplasm (*n* = 15) were the most frequent other hematologic malignancies (*n* = 45). In comparison to patients with s-AML following MDS, those with s-AML following other malignancies were more frequently ≤45 years old (42.5% vs. 26.5%), more frequently female (60% vs. 40%), had longer latency time to develop leukemia from their primary disease (median 3.7 vs. 0.6 years) and had more frequently good risk cytogenetics (10% vs. 0.5%). Consequently, they were more likely to achieve a complete remission (CR) or complete remission with incomplete count recovery (CRi) after one or two courses of induction chemotherapy (65% vs. 51%), had a lower relapse rate (37% vs. 58%) and had a higher 5-year OS rate (34% vs. 21%) (Table 1).

#### 2.2.2. Factors Affecting OS

We first assessed whether there was an improvement of OS duration according to the start of treatment: before 1990, from 1990 through 1999 or from 2000 onward. As shown in Figure 1A, there was a systematic improvement in 5-year OS rate over the three time periods for patients in both groups A and B, as well as in each subgroup: A (0% vs. 18% vs. 27%, *p* = 0.0004) and B (12% vs. 29% vs. 45%, *p* = 0.007) (data not shown).

Previous studies have suggested that younger patients (age ≤ 45 years) have better outcomes than older patients following intensive induction chemotherapy. Generally, before mid-1980s, allo-SCT was offered in EORTC collaborative studies only to younger patients. We compared OS in patients ≤ or >45 years of age. We observed that younger patients (16-45 years old) had a significantly longer OS compared to older patients (Figure 1B). This was true both in group A (37% vs. 15%) and in group B (44% vs. 27%). Furthermore, interestingly, female patients had a sustainable longer OS than males (5-year OS rate: 36% vs. 19%; HR = 0.67; *p* = 0.001) As expected, those with a WHO PS 2, 3 or 4 had, initially, a higher death rate than those with a WHO PS 0 or 1; however, the 5-year OS rates were identical (28%) (HR = 1.30; *p* = 0.10).

High WBC at diagnosis is a well-known risk factor in de novo AML. In the current analysis, we observed that s-AML patients with WBC ≥25 × 10^9^/L had a 5-year OS rate of 19% (95% CI: 12–28%) versus 31% (95% CI: 25–37%) in those with <25 × 10^9^/L (HR = 1.60; *p* < 0.001).

Similar to what has been observed for de novo AML, s-AML patients with good or intermediate risk cytogenetics had a better OS than those with poor-risk cytogenetics (Figure 1C). Furthermore, as recently reported in all AML patients included in EORTC/GIMEMA AML-10 and AML-12 trials [8], patients with a monosomal karyotype (MK+, *n* = 23) in the current study had a very poor outcome (0% OS rate at 2.5 years versus 30% OS rate at 5 years for those without a MK) (*p* < 0.0001, Figure 1D).

In addition, we assessed the impact of the dose of cytarabine (AraC) in the induction treatment on OS. Patients treated with high-dose (HD)-AraC were all enrolled in the EORTC AML-12 randomized trial [9]. We observed that patients given HD-AraC had longer OS (HR = 0.44) than those given the standard dose (SD)-AraC (Figure 1E). The beneficial impact of HD-AraC remained present when restricting the analyses to patients starting the induction treatment ≥2000 (Figure 1F) or to patients included in the AML-12 study. As shown in the Appendix A, the benefit of HD-AraC on OS was particularly more important (test for interaction: *p* = 0.06) in patients <46 years (HR = 0.21) than in those 46–60 years old (HR = 0.61), but was consistent according to sex, WBC at diagnosis, cytogenetic risk group and cohort (A versus B).

In a Cox multivariate analysis adjusted by cohort (A versus B), male (HR = 1.39, *p* = 0.01), WBC ≥25 × 10^9^/L (HR = 2.00; *p* < 0.0001), age 46–60 years (HR = 1.65; *p* < 0.001) and poor-risk cytogenetics (HR = 2.17, *p* < 0.0001) were independently associated with a shorter OS while being treated between 1990 and 1999 (HR = 0.50, *p* = 0.003) or ≥ 2000 (HR = 0.43, *p* = 0.0008), and having received HD-AraC in the induction-remission chemotherapy (HR = 0.54, *p* = 0.012) were each associated with a longer OS (Table 2, model 1). The improvement of OS due to HD-Ara-C was particularly striking in patients <46 years (HR = 0.17) (Table 2, model 2).

#### 2.2.3. Impact of allo-SCT and auto-SCT on OS

Forty-six of the 213 patients who achieved a CR/CRi received an allo-SCT in first CR/CRi. Their 5-year OS rate from allo-SCT was 65% (95% CI: 50–77%). Interestingly, it was similar in patients treated < or ≥2000. Another 57 patients underwent an auto-SCT in first CR/CRi. Their 5-year OS rate from the time of auto-SCT was 55% (95% CI: 41–67%). As expected, the nonrelapse mortality in the allo-SCT group was higher than in the auto-SCT group (24% vs. 11%), but the relapse rate was much lower (11% vs. 40%). In a Cox model for OS in which allo-SCT and auto-SCT in CR1 were modeled as time-dependent variables, the HR was 0.76 (95% CI: 0.40–1.44) in favor of the allo-SCT group.

### 2.3. Outcomes in Patients > 60 Years Old with s-AML Occurring after MDS or Other Malignancies

#### 2.3.1. General Characteristics and Outcomes

Baseline characteristics of older patients (*n* = 502), with s-AML following either MDS (cohort A, *n* = 327) or other malignancies (cohort B, *n* = 175) are summarized in Table 3. Among other malignancies, breast (*n* = 32), gastro-intestinal (*n* = 19), genitourinary (*n* = 17), gynecologic (*n* = 13), head and neck (*n* = 13) and skin (*n* = 6) cancers were the most frequent solid tumors (*n* = 132), while 36 patients had s-AML following other hematologic malignancies. In comparison with patients with s-AML following MDS, those with s-AML following other malignancies were more frequently females (51% vs. 40%), had longer latency time to develop leukemia from their primary disease (median 6.4 vs. 0.65 years) and had less frequently a monosomal karyotype (5% vs. 12%). Therefore, the CR/CRi rate was numerically higher in cohort B than in cohort A (49% vs. 42%) while the 5-year OS rate was dismal in both arms (5% vs. 9% *p* = 0.6) (Table 3).

#### 2.3.2. Factors Associated with OS

While patients who started the treatment <1990 had a lower 5-year OS rate than those who started ≥1990 (3% vs. 7%, *p* = 0.02), there was no improvement of OS since 1990 (Figure 2A). Furthermore, interestingly, patients <70 years did not have better outcomes than older patients (*p* = 0.81), while OS was superimposable in male and female patients (*p* = 0.70). However, WHO-PS strongly correlated with OS (median OS of 0.8 years in patients with WHO PS 0-1 versus 0.4 years of for those with PS 2–4) (*p* < 0.001, Figure 2B).

As observed in younger patients, there was a strong impact of cytogenetic risks on OS (*p* = 0.0001, Figure 2C). Furthermore, patients with a monosomal karyotype had a particularly poor outcome (median OS 0.3 years vs. 0.9 years for patients without a monosomal karyotype) (*p* < 0.0001, Figure 2D). Finally, median OS decreased as WBC × 10^9^/L at diagnosis increased (0.9 years for patients with WBC <2.5, vs. 0.7 years for patients with WBC between 2.5 and 24.99, vs. 0.5 years for patients with WBC ≥25) (*p* = 0.016).

A total of 136 s-AML patients were randomized between standard induction without (*n* = 67) or with (*n* = 69) pre-treatment with GO in the AML-17 EORTC/GIMEMA protocol [10]. Overall, GO treatment had no impact on the OS (HR = 0.91, *p* = 0.59). However, as observed in the general population of the protocol, there was an interaction between age and the randomized group: in patients <70 years (*n* = 84), GO-treated patients had an OS that was similar to those in the standard group (HR = 0.73) (Figure 2E), while for patients ≥70 years (*n* = 52), the OS was shorter (median 0.4 vs. 0.7 years) (Figure 2F).

In a multivariate Cox model, presence of MK or adverse cytogenetics, WHO PS 2–4, WBC >25 × 10^9^/L and treatment start before 1990 were associated with a shorter OS.

## 3. Discussion

Most current knowledge on s-AML is derived from retrospective studies with relatively low numbers of patients, from National Cancer Registries [7], or from transplantation registries [11,12]. Here, we report data from an EORTC database that includes data from s-AML patients who were treated in 13 EORTC collaborative trials conducted between May 1986 and January 2008. Several observations were made.

A first observation was that the incidence of s-AML among patients included in the 13 EORTC collaborative trials was 11%. This is somewhat lower than what has been reported in most recent National Cancer Registries in which s-AML represent up to 25% of all AML cases [3,7]. This is probably the reflection of the eligibility criteria in the various clinical trials.

A second observation was that there was a significant improvement over time in OS in s-AML patients <60 years of age at inclusion. This is consistent with what has been observed in patients with de novo AML [13]. This could probably be mostly attributed to improvement in supportive care (such as the introduction of newer antibiotics and of newer antifungal medication active against molds and safer transfusion practices) as well as to impressive improvements in the field of allo-SCT [14,15].

Another important observation of our study was a significantly better OS in s-AML patients receiving HD-AraC as induction chemotherapy. This observation remained true when restricting the analysis to patients who were included in the AML-12 trial in which patients were randomized between SD-AraC and HD-AraC. While a better OS in patients receiving HD-AraC was observed in AML-12 in the subgroup of patients <46 years of age [9], the patients with s-AML appeared to benefit even more than those with de novo AML. Interestingly, in our study, focused on s-AML patients, whether included in the AML-12 study or not, we could confirm that patients 15–45 years old had benefited the most of HD-AraC, more than those 46–60 years old. These results could be the basis of a new trial assessing HD-AraC in younger s-AML patients, perhaps excluding patients with a MK since we previously observed that MK AML patients do not benefit from HD-AraC [8]. It should be noted that patients included in the AML-12 received a dose of daunorubicin of 50mg/m^2^ on days 1, 3 and 5, which is nowadays considered inadequately low (although all patients received 5 days of etoposide 50 mg/m^2^ in addition). Whether the benefit of HD-AraC will remain present in s-AML patients receiving 60 mg/m^2^ of daunorubicin three times should also be investigated. Interestingly, our observation is in line with those recently reported by Vulaj et al., who observed better response rates with FLAG (fludarabine, high-dose cytarabine and G-CSF) than with 3 + 7 in a retrospective study including sAML patients [16].

A fourth observation of our study was a suggestion for better OS in s-AML patients receiving an allo-SCT than those given an auto-SCT (HR 0.76, 95% CI: 0.40–1.44). The difference did not reach statistical significance, but this comparison had a low statistical power given the low number of patients in both groups. The longer OS after an allo-SCT would be consistent with what has been observed in the overall AML population without good-risk cytogenetics and with the recent demonstration of a graft-versus-leukemia effect in patients with s-AML [17]. Five-year OS rate from transplantation in patients receiving an allo-SCT was 65% (95% CI: 50–77%) in our study. This is somewhat better than what has been observed in the EBMT registry where 2-year OS in s-AML patients in CR was 56% (95% CI: 54–59%) [17].

In contrast to what has been observed in s-AML patients <60 year of age, we failed to observe a similar improvement with time in older s-AML patients. This could be at least partly explained by the fact that these patients were generally not offered an allo-SCT, since several advances, such as the development of CPX-351, azacitidine or decitabine maintenance after intensive chemotherapy or the development of allo-SCT following reduced-intensity or truly non-myeloablative conditioning, are likely to result in OS improvement in s-AML patients [18,19,20].

This study has some limitations: because of its inherent retrospective nature, it comprised patients included in different multicenter trials, and thus, a certain heterogeneity of treatment received was present. Moreover, the series of patients treated before 1990 was limited, cytogenetic data were missing for a substantial proportion (approximately 30%) of patients and molecular data were lacking. Additionally, it is likely that a selection bias occurred, as the vast majority of the patients had an ECOG PS of 0 or 1, which is probably not the case in “real life” patients, treated outside prospective clinical trials. Finally, it should be stressed that most recent patients included in this retrospective study were included 12 years ago.

## 4. Patients and Methods

### 4.1. Patients, Treatment, Data Management and Methods

We explored the EORTC- leukemia group (EORTC-LG) trial database for s-AML patients in order to evaluate the outcome of s-AML over time and to gain insight on the characteristics of this disease.

Between 1986 and 2008, 16 consecutive collaborative trials were conducted by the Adult Leukemia Groups of EORTC, GIMEMA and HOVON. We selected 13 trials by their availability of electronic data; one trial investigated consolidation therapy, enrolling only patients who were already in CR or CRi, which was thus not selected. The selected trials included EORTC protocol numbers 06012 [21], 06013 [22], 06862 [23], 06863 [24,25], 06864 [26], 06888 [27], 06892 [28], 06921 [29], 06931 [30], 06954 [31], 06961 [32], 06991 (AML-12) [9] and 06993 [10] (Appendix A).

All patients received an intensive induction chemotherapy including cytarabine in a standard (SD-AraC) or high dose (HD-Ara-C; i.e., only patients included in the AML-12 trial) and an anthracycline. Patients included in the 06991 study (AML-12) were randomized in a group with SD-AraC induction, namely DNR (50 mg/m^2^ per day on days 1, 3 and 5), plus etoposide (50 mg/m^2^ per day on days 1-5) plus 10 days of cytarabine (100 mg/m^2^ per day as continuous IV infusion) or in a group with HD-AraC induction: DNR (50 mg/m^2^ on days 1, 3 and 5), plus etoposide (50 mg/m^2^ per day on days 1-5) plus cytarabine (3000 mg/m^2^ every 12 h as a 3-h IV infusion on days 1, 3, 5 and 7) [9]. Patients included in the 06012 study (AML17) were randomized between induction chemotherapy with mitoxantrone, SD-AraC and etoposide preceded, or not, by a course of GO (6 mg/m^2^ on days 1 and 15) [21]. Patients in CR or in CR with incomplete recovery (CRi) (generally assessed by day 31 of induction) received two consolidation courses with or without GO (3 mg/m^2^ on day 0). To ensure homogeneity in the documentation of data, new case record forms (CRFs) were constructed in light of the most updated classification as of August 2013 of the specific parameters whenever applicable. A new database was created, and populated with the data extracted from the 13 selected trials, after an additional data cleaning.

AML diagnosis was confirmed according to the 2008 WHO classification with bone marrow (BM), or peripheral blood blasts ≥20% (the vast majority of patients where included in protocols which used the cutoff of ≥30% since they were developed before 2008). Patients were classified as s-AML, if they had: (1) a documented prior malignant or non-malignant disease with a history of cytotoxic or immune suppressive treatment or any known toxic exposure whether for therapeutic, environmental or occupational reason, or (2) patient files reviewed and confirmed by study coordinator at time of enrollment as s-AML. AML-M3 and MDS without confirmed diagnosis of > 2 months before AML development were excluded according to the selection criteria used by the EORTC, GIMEMA and HOVON-LG groups to define s-AML following MDS in these studies.

Cytogenetic data were reviewed and recoded according to the ISCN 2013 and cytogenetic risk groups determined based on the refined UK Medical Research Council (MRC) prognosis classification criteria [33]. Monosomal karyotype (MK) was defined as at least two autosomal monosomies or one single autosomal monosomy in combination with at least one structural abnormality [34,35].

### 4.2. Statistical Analysis

Patients were divided into three cohorts, cohort A (s-AML after MDS), cohort B (s-AML after other malignant diseases) and cohort C (after non-malignant conditions or toxic exposures). Analyses were performed for two age groups: younger patients, aged between 16 to 60 years at inclusion in their respective clinical trial, and older patients, aged 61 to 85 years. Overall survival (OS) was calculated from the date of start of treatment until death, irrespective of the cause. Disease-free survival (DFS) was calculated as the time from CR/CRi until the first relapse or death, irrespective of the cause. The Kaplan-Meier method was used to estimate OS and DFS distributions, and the logrank test to compare them. Uni- and multivariable analyses were performed using Cox proportional hazards model [36]. SAS 9.4 software, Cary, NC, USA was used for the statistical analyses.

## 5. Conclusions

In summary, we report here the results of a large cohort of patients with s-AML included in clinical trials over a 22-year period. We observed that the outcome of s-AML in younger patients improved over the years in parallel with introduction of HD-AraC in induction remission chemotherapy. This suggests that HD-AraC could be considered for younger patients with s-AML. Furthermore, this study confirmed that the sAML patients having adverse cytogenetic risk features and/or a monosomal karyotype, and those with high WBC (≥25 × 10^9^/L) at diagnosis had a dismal OS outcome, regardless of their age group.

## Figures and Tables

**Figure 1 cancers-12-03334-f001:**
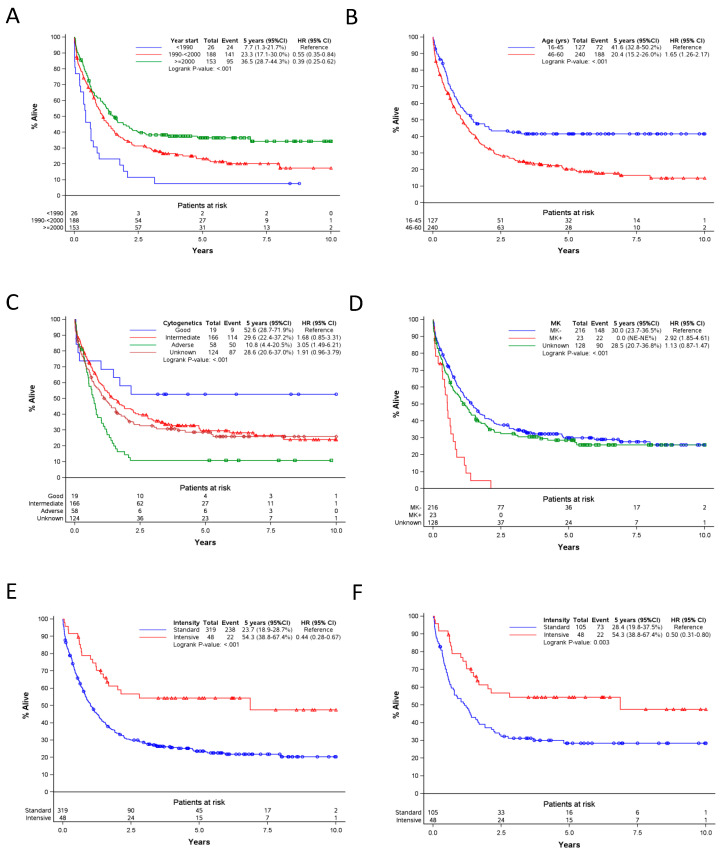
(**A**,**B**), in younger patients (age ≤ 60 years): association between treatment period (**A**), initial features (**B**–**D**), treatment intensity (high dose versus standard dose AraC) ((**E**,**F**) restricted to patients treated ≥2000) and OS.

**Figure 2 cancers-12-03334-f002:**
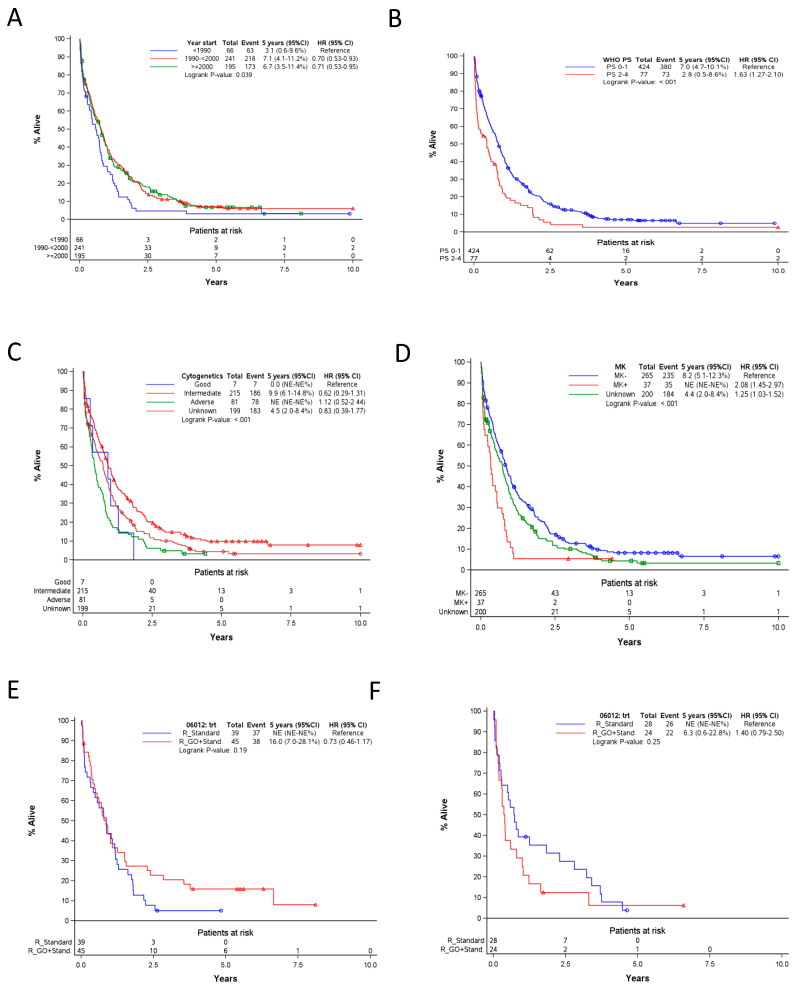
(**A**,**B**) in older patients (age > 60 years): association between treatment period (**A**), initial features (**B**–**D**), treatment intensity (GO or not) ((**E**,**F**) restricted to patients >70 years of age) and OS. GO, Gemtuzumab ozogamicin; R_GO+Stand, randomized to receive GO and standard treatment; R_Standard, randomized to receive standard treatment

**Table 1 cancers-12-03334-t001:** Baseline characteristics, treatment period and intensity, autologous stem cell transplantation (auto-SCT) and allo-SCT applicability and outcomes in patients with s-AML following MDS (group A) or other malignancies (group B), aged 16–60 years at the start of treatment.

Variation	Group		All	Age	
	Group A (*n* = 181)	Group B (*n* = 186)	Total (*n* = 367)	16–45 yrs (*n* = 127)	46–60 yrs (*n* = 240)
Year start of treatment; *n* (%)					
<1990	10 (5.5)	16 (8.6)	26 (7.1)	13 (10.2)	13 (5.4)
1990–<2000	98 (54.1)	90 (48.4)	188 (51.2)	64 (50.4)	124 (51.7)
≥2000	73 (40.3)	80 (43.0)	153 (41.7)	50 (39.4)	103 (42.9)
Sex, *n* (%)					
Male	109 (60.2)	74 (39.8)	183 (49.9)	65 (51.2)	118 (49.2)
Female	72 (39.8)	112 (60.2)	184 (50.1)	62 (48.8)	122 (50.8)
Age at Start of Treatment (years)					
Median (range)	51 (20–60)	48 (16–60)	49 (16–60)	39.0 (16.0-45.0)	54.0 (46.0-60.0)
16-45, *n* (%)	48 (26.5)	79 (42.5)	127 (34.6)	127 (100.0)	
46-60, *n* (%)	133 (73.5)	107 (57.5)	240 (65.4)		240 (100.0)
WHO PS, *n* (%)					
0	78 (43.1)	69 (37.1)	147 (40.1)	43 (33.9)	104 (43.3)
1	71 (39.2)	80 (43.0)	151 (41.1)	61 (48.0)	90 (37.5)
2, 3 or 4	32 (17.7)	36 (19.4)	68 (18.5)	22 (17.3)	46 (19.2)
WBC at diagnosis (×10^9^/L)					
Median (range)	5.3 (0.6–408.3)	9.6 (0.5–290.0)	6.8 (0.5–408.3)	39.0 (16.0–45.0)	54.0 (46.0–60.0)
<25	140 (77.3)	120 (64.5)	260 (70.8)	85 (66.9)	175 (72.9)
≥25	41 (22.7)	65 (34.9)	106 (28.9)	41 (32.3)	65 (27.1)
Cytogenetics Risk Group, *n* (%)					
Good	1 (0.6)	18 (9.7)	19 (5.2)	11 (8.7)	8 (3.3)
Intermediate	87 (48.1)	79 (42.5)	166 (45.2)	46 (36.2)	120 (50.0)
Adverse	36 (19.9)	22 (11.8)	58 (15.8)	20 (15.7)	38 (15.8)
Unknown	57 (31.5)	67 (36.0)	124 (33.8)	50 (39.4)	74 (30.8)
Monosomal Karyotype (MK) Status, *n* (%)					
MK−	108 (59.7)	108 (58.1)	216 (58.9)	68 (53.5)	148 (61.7)
MK+	14 (7.7)	9 (4.8)	23 (6.3)	7 (5.5)	16 (6.7)
Unknown	59 (32.6)	69 (37.1)		52 (40.9)	76 (31.7)
Time From MDS to AML (months)	*n* = 127	*n* = 13	*n* = 140	*n* = 37	*n* = 103
Median (range)	7.2 (2.1–178.4)	5.3	7.1 (0.4–178.4)	5.3 (0.7–1.78.4)	8.0 (0.4–139.7)
Time From Other Cancer to AML (years)					
Median (range)	NA	3.7 (0.2–30.3)	3.7 (0.2–30.3)	3.1 (0.2–30.3)	4.4 (0.8–22.8)
Induction Treatment Intensity					
Standard	163 (90.1)	156 (83.9)	319 (86.9)	107 (84.3)	212 (88.3)
Intensive	18 (9.9)	30 (16.1)	48 (13.1)	20 (15.7)	28 (11.7)
Best Response to Induction Treatment, *n* (%)					
No CR/CRi	87 (49.2)	65 (34.9)	154 (42.0)	41 (32.3)	113 (47.1)
CR/CRi	92 (50.8)	121 (65.0)	213 (58.0)	86 (67.7)	127 (53.0))
Auto SCT in CR/CRi, *n* [%]	17 [18.5]	40 [33.1]	57 [26.8]	28 [32.6]	29 [22.8]
Allo SCT in CR/CRi, *n* [%]	23 [25.0]	23 [19.0]	46 [21.6]	26 [30.2]	20 [15.7]
DFS Status in CR/CRi Patients, *n* [%]					
Still in First CR/CRi	29 [31.5]	53 [43.8]	82 38.5]	48 [55.8]	34 [26.8]
Relapse after CR/CRi	53 [57.6]	45 [37.2]	98 [46.0]	28 [32.6]	70 [55.1]
Death in CR/CRi	10 [10.9]	23 [19.0]	33 [15.5]	10 [11.6]	23 [18.1]
Survival Status, *n* (%)					
Alive	42 (23.2)	65 (34.9)	107 (29.2)	55 (43.3)	52 (21.7)
Dead	139 (76.8)	121 (65.1)	260 (70.8)	72 (56.7)	188 (78.3)

Percentages between [ ] were computed among patients who reached CR/CRi. WHO PS, World Health Organization performance status scale; DFS, disease-free survival.

**Table 2 cancers-12-03334-t002:** Results of the multivariable Cox model for OS in patients with s-AML following MDS (group A) or other malignancies (group B) and aged 16–60 years at the start of treatment.

Variable	Category	Hazard Ratio (HR)	95%CI for HR	*p*-Value *
Model 1				
Group	Group A	1		
	Group B	0.84	(0.65, 1.09)	0.20
Sex	Male	1		
	Female	0.72	(0.56, 0.93)	0.01
WBC (×10^9^/L)	<25	1		
	≥25	1.99	(1.51, 2.62)	<0.001
Cytogenetics	Good/intermediate	1		(<0.001)
	Adverse	2.17	(1.54, 3.06)	<0.0001
	Unknown	1.25	(0.93, 1.67)	0.14
Period	<1990	1		(0.005)
	1990 ≤ 2000	0.50	(0.31, 0.80)	0.003
	≥2000	0.43	(0.26, 0.70)	0.0008
Age (years)	16–45	1		
	46–60	1.65	(1.24, 2.18)	0.0005
Treatment intensity	Standard	1		
	Intensive	0.54	(0.33, 0.87)	0.01
Model 2				
Group	Group A	1		
	Group B	0.82	(0.63, 1.06)	0.13
Sex	Male	1		
	Female	0.68	(0.53, 0.87)	0.003
WBC (×10^9^/L)	<25	1		
	≥25	2.14	(1.62, 2.82)	< 0.001
Cytogenetics	Good/intermediate	1		(<0.001)
	Adverse	2.22	(1.57, 3.12)	<0.0001
	Unknown	1.27	(0.95, 1.70)	0.11
Period	<1990	1		(0.005)
	1990–<2000	0.51	(0.32, 0.81)	0.004
	≥2000	0.44	(0.27, 0.72)	0.001
Age/treatment intensity	16–45 y/Standard	1		(0.001)
	16–45 y/Intensive	0.17	(0.06, 0.49)	0.001
	46–60 y/Standard	1.41	(1.06, 1.89)	0.019
	46–60 y/Intensive	1.28	(0.72, 2.25)	0.40

*: for the overall comparison, between brackets; for the pairwise comparison, with the baseline category. If, instead of variable “age/treatment intensity,” one includes categorical variables for age and treatment intensity, the results of the Cox model were, for age 46–60 vs. 16–45 years, HR = 1.65, 95% CI: 1.24–2.18, and for intensive vs. standard HR = 0.54, 95% CI: 0.33–0.87, but the prognostic importance of the model was lower than the one indicated in Model 2.

**Table 3 cancers-12-03334-t003:** Baseline characteristics, treatment period and outcomes in patients with s-AML following MDS (group A) or other malignancies (group B), and for those older than 60 years at the start of treatment.

Variable	Group A (*n* = 327)	Group B (*n* = 175)	Total (*n* = 502)
Year start, *n* (%)			
<1990	38 (11.6)	28 (16.0)	66 (13.1)
1990 ≤ 2000	149 (45.6)	92 (52.6)	241 (48.0)
≥2000	140 (42.8)	55 (31.4)	195 (38.8)
Sex, n (%)			
Male	195 (59.6)	85 (48.6)	280 (55.8)
Female	132 (40.4)	90 (51.4)	222 (44.2)
Age at Start of Induction			
Median (range), years	68.0 (61.0–85.0)	68.0 (61.0–85.0)	68.0 (61.0–85.0)
61–69, *n* (%)	191 (58.4)	115 (65.7)	306 (61.0)
70–85, *n* (%)	136 (41.6)	60 (34.3)	196 (39.0)
WHO PS, *n* (%)			
PS 0	121 (37.0)	54 (30.9)	175 (34.9)
PS 1	161 (49.2)	88 (50.3)	249 (49.6)
PS 2, 3 or 4	45 (13.7)	32 (18.3)	77 (15.3)
WBCx10^9^/L at diagnosis, *n* (%)			
<25	250 (76.5)	119 (68.0)	369 (73.5)
25 ≤ 100	65 (19.9)	41 (23.4)	106 (21.1)
≥100	12 (3.7)	15 (8.6)	27 (5.4)
Cytogenetics, *n* (%)			
Good	1 (0.3)	6 (3.4)	7 (1.4)
Intermediate	151 (46.2)	64 (36.6)	215 (42.8)
Adverse	50 (15.3)	31 (17.7)	81 (16.1)
Unknown	125 (38.2)	74 (42.3)	199 (39.6)
Monosomal Karyotype (MK) status, n (%)			
MK−	186 (56.9)	79 (45.1)	265 (52.8)
MK+	16 (4.9)	21 (12.0)	37 (7.4)
Unknown	125 (38.2)	75 (42.9)	200 (39.8)
Time from MDS to AML (months)			
Median (range)	7.8 (2.1–126.0)	11.3 (1.1–58.2)	8.2 (1.1–126.0)
Time from Other Cancer to AML (years)			
Median (range)		6.4 (0.1–38.0)	6.4 (0.1–38.0)
CR/CRi Status after Induction, *n* (%)			
No CR/CRi	188 (57.5)	90 (51.4)	278 (54.4)
CR/CRi	139 (42.5)	85 (48.6)	224 (44.6)
DFS Status in Patients in CR/CRi, *n* [%]			
Still in CR/CRi	11 [7.9]	10 [11.8]	21 [9.4]
Relapse after CR/CRi	115 [82.7]	62 [72.9]	177 [79.0]
Death in CR/CRi	13 [9.4]	13 [15.3]	26 [11.6]
Survival Status, n (%)			
Alive	27 (8.3)	21 (12.0)	48 (9.6)
Dead	300 (91.7)	154 (88.0)	454 (90.4)
5-year Survival Rate (95% CI)	5% (3–8%)	9% (5–14%)	6% (4–9%)

Percentages between [ ] were computed among patients who reached CR/CRi.

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
