# Peer review of "Survival Improvement over Time of 960 s-AML Patients Included in 13 EORTC-GIMEMA-HOVON Trials"

_cancers, 2020, doi:10.3390/cancers12113334_

Round 1
Reviewer 1 Report
Dear Editors,
Overall I think this is a very good study and recommend Minor Revisions.
Study highlights
- Very large cohort of patients allowing to analyze this rare entity on a large number of patients, which makes the statistics quite consistent. However, caution should be done in the analysis and conclusion of the results, given the retrospective nature and the fact that these are different timings and treatments.
- Long period follow up: almost 6 years.
- Rate of sAML, either tAML or AML-MRC comparable to known data.
- Good balance between the groups in terms of number of patients, age, follow up, causes.
- Results fairly consistent with the literature data regarding outcomes in the 2 groups (post treatment and post MDS) in both age cohorts.
Weaknesses
- The reader has a lot of difficulty to find his way through the numerous data described, what is the key message? Description of patient population ? Effectiveness over time? Impact of treatment? Age-related differences in outcomes? Maybe too much information.
- Despite large numbers of patients, this is still a retrospective analysis which should be interpreted with caution, especially with regard to the quite different treatment groups.
-These are fairly old data not following the current classification, WHO 2016, including t-AML and AML-MRC. The distribution into 3 distinct groups A, B, C is not entirely consistent in this framework, it should be noted.
- Too few patients treated before 1990 compared to other periods for the analysis to have sufficient statistical weight.
- Beware of the impact of aracytin HD which is not strongly confirmed statistically, HR 0.54, P0.012. These data are also difficult to interpret given the small number of patients treated in the HD group. Can a possible effect of ECOG or cytogenetics be excluded?
- Possible selection bias, majority of ECOG 0 and 1 in the two different age cohorts.
- Large number of unknown cytogenetic results (more than 30% in all groups), which makes the interpretation of the results difficult given the major prognostic impact of cytogenetics, also demonstrated in this study. Once again, is the better prognosis observed in the young patient population correlated with cytogenetic abnormalities as expected?
How to explain that 5-year survival post allo = post auto. Place of auto not well defined in these patient groups. This would justify a comment by the author.
- Interest of the comparison of these results with other AML ? With known data ?
Conclusion
It is an interesting work on a little studied disease, with the great advantage of the number of patients. In my opinion, what is really consistent in the multivariate analyses and should be the subject of the conclusion is the incidence of cases in the groups, the improving trend over the years, the significant impact of cytogenetics, ECOG and WBC count as we know it. The place of HD cytosar and the impact of age in the subgroups should be considered with much more caution due to differences between groups, small numbers of patients and statistical data. The authors should emphasize the importance of validating these results in prospective studies. On the other hand, the <60 and >60 year curves are very different and should be highlighted. The authors should chose and emphasize the key messages because the reader may be lost in too much information. The retrospective nature of the data over a very long period of time should be underlined whit probably a certain bias related to inclusion in studies, again suggesting the need of prospective validation.
Author Response
Overall I think this is a very good study and recommend Minor Revisions.
Re. We would like to thank the reviewer for these positive comments.
Study highlights
- Very large cohort of patients allowing to analyze this rare entity on a large number of patients, which makes the statistics quite consistent. However, caution should be done in the analysis and conclusion of the results, given the retrospective nature and the fact that these are different timings and treatments.
- Long period follow up: almost 6 years.
- Rate of sAML, either tAML or AML-MRC comparable to known data.
- Good balance between the groups in terms of number of patients, age, follow up, causes.
- Results fairly consistent with the literature data regarding outcomes in the 2 groups (post treatment and post MDS) in both age cohorts.
Weaknesses
- The reader has a lot of difficulty to find his way through the numerous data described, what is the key message? Description of patient population? Effectiveness over time? Impact of treatment? Age-related differences in outcomes? Maybe too much information.
Re. We agree that the manuscript includes a lot of information. We tried to help the reader to identify the key messages with a clear title, the description of the main findings in the abstract and by showing two comprehensive figures summarizing the more important findings of the study. We have also added a Conclusions paragraph summarizing the main findings of the manuscript.
- Despite large numbers of patients, this is still a retrospective analysis which should be interpreted with caution, especially with regard to the quite different treatment groups.
Re. We thank the reviewer for this comment. We agree and have outlined these limitations in the discussion part of the manuscript. See manuscript lines 255-256.
-These are fairly old data not following the current classification, WHO 2016, including t-AML and AML-MRC. The distribution into 3 distinct groups A, B, C is not entirely consistent in this framework, it should be noted.
Re. Every classification is changing, but the main criteria remained the same. For the AML genetic classification, we reported in the manuscript and analyzed with the MRC definition (see lines 297-298). Regarding the % of BM blasts, we generally used the 30% (old) cut-point as mentioned in the protocol (see manuscript lines 289-290).
- Too few patients treated before 1990 compared to other periods for the analysis to have sufficient statistical weight.
Re. We agree and have added this in the limitation section of the manuscript. See manuscript lines 257-258.
- Beware of the impact of aracytin HD which is not strongly confirmed statistically, HR 0.54, P=0.012. These data are also difficult to interpret given the small number of patients treated in the HD group. Can a possible effect of ECOG or cytogenetics be excluded?
Re. We thank the reviewer for this comment. Indeed the benefit for HD-araC remains statistically present when adjusted for cytogenetics in Cox model (see table 2). This effect was more present in the subgroup of younger patients (16-45 years of age) with an HR=0.17, P=0.001. We have outlined this information in the discussion part of the manuscript. See manuscript lines 228-230.
- Possible selection bias, majority of ECOG 0 and 1 in the two different age cohorts.
We agree with the reviewer and have added this limitation to the manuscript (see lines 258-260).
- Large number of unknown cytogenetic results (more than 30% in all groups), which makes the interpretation of the results difficult given the major prognostic impact of cytogenetics, also demonstrated in this study. Once again, is the better prognosis observed in the young patient population correlated with cytogenetic abnormalities as expected?
Re. We thank the reviewer for these comments. As in many older series, cytogenetic analysis could not be successfully be performed, due to too few mitoses, etc. Over time, the proportion of patients with an inevaluable cytogetic exam remained quite stable over time in the EORTC studies. As indicated in the Table 1, this proportion was very similar in the 2 groups: Group A (31.5%) and Group B (36.0%). In our multivariable analyses, for variable “cytogenetics” we did consider a specific category “Unknown”, for those patients with an inevaluable cytogenetic risk group. In this way, we allowed to perform the multivariable analysis on the same entire cohort of patients. In this way, we could assess the prognostic importance of all variables in the multivariate setting, allowing the other variables to express their prognostic importance relative to the 3 cytogenetic risk groups: “Good/intermediate”, “Poor”, “Unknown”. The Indeed, as expected, the proportion of patients with good risk cytogenetic was indeed higher in patients aged 15-46 years of age.
How to explain that 5-year survival post allo = post auto. Place of auto not well defined in these patient groups. This would justify a comment by the author.
Re. There was indeed not significant differences between AlloSCT and AutoSCT in our cohort of patients. Specifically, in a Cox model for OS from SCT in CR1, in which AlloSCT and AutoSCT in CR1 were considered as time-dependent variables, the HR was 0.76 (95% CI: 0.40-1.44). Given the number of transplanted patients there was little power to detect significant differences between AlloSCT and AutoSCT. We have added the results of the Cox model in the result part of the manuscript (see manuscript lines 165-166). We have also discussed these findings in the discussion part of the manuscript (see manuscript lines 240-243).
- Interest of the comparison of these results with other AML? With known data ?
Re. We agree with the reviewer that this would have been of interest. However, this analysis is not possible since the new database included only data from sAML patients.
Finally, we would like to thank very much the reviewer for his constructive criticisms that helped us improving our manuscript.
Reviewer 2 Report
This is important very large retrospective analysis of 13 European coop groups trials (EORTC-GIMEMA-HOVON) including 8,858 pts of which 960 had sAML. Study is well done methodologically, well written, and presents important findings.
There are some very interesting findings here in one of the largest studies ever looking at secondary AML.
Authors show that Survival has improved over last 3 decades in younger but not older adults with secondary AML (sAML). There is also a suggestion that HiDAC with induction may benefit these pts as well, i.e. not just in de novo or fav/int risk as has been prev published.
- It would be nice to see more pts from after 2000 (this analysis stops at 2008). However, this may be a function of when these trials were done, follow-up time, etc.
- It is interesting that on AML-12 study pts w sAML also had better OS with HiDAC-based induction. In most places, SD-araC is still used, but for sAML or otherwise high risk AML a day 14 marrow would be done and if positive without substantial reduction in blasts – then HiDAC may then be given. Was this common practice on these trials that did not include HiDAC upfront with the initial induction? And if so, is this any different than giving it initially?
- Some of the trials included etoposide or mitoxantrone, or GO, which is a bit confounding. However, this is going to be an issue with any large retrospective study combining trials.
- Total and in either age groups, <=60 and >60, was there any difference in outcome if pts had sAML from MDS (without prior therapy) vs sAML from prior chemo/rads? I noted there were more fav risk and less adverse in therapy related AML group, which at least in part makes sense biologically as some of the translocations with be CBF etc.
- The fact that females has much better OS is very interesting and could be further explored. Also, not unexpected but notable that higher WBC associated with worse survival in sAML, as this usually more associated with de novo disease.
- The 5-yr OS rates post allo and post AUTO HSCT seem quite good. Were they just all younger pts? Any more to be made of this, esp the auto-transplant data?
- Was there any difference in outcome among the t-AML pts based on the prior malignancy type? Or for prior heme vs. solid tumor malignancy?
- Finally, the 5-yr OS rate was much worse in over 60, not unsurprising. How many of these pts were transplanted and does this in part explain this? Possibly this reflects that fewer older pts were transplanted in the eras that these trials were done. Biologically the disease “should” be the same, so is this just driven by worse PS/organ function and less application of allo-BMT, or is there some other biologic/epigenetic difference between over 60 and under 60, even if they both have sAML?
Author Response
This is important very large retrospective analysis of 13 European coop groups trials (EORTC-GIMEMA-HOVON) including 8,858 pts of which 960 had sAML. Study is well done methodologically, well written, and presents important findings.
There are some very interesting findings here in one of the largest studies ever looking at secondary AML.
Authors show that Survival has improved over last 3 decades in younger but not older adults with secondary AML (sAML). There is also a suggestion that HiDAC with induction may benefit these pts as well, i.e. not just in de novo or fav/int risk as has been prev published.
Re. We thank the reviewer for these positive comments.
- It would be nice to see more pts from after 2000 (this analysis stops at 2008). However, this may be a function of when these trials were done, follow-up time, etc.
Re. We thank the reviewer for this comment. Indeed, unfortunately no patients treated after 2008 were included in this study.
- It is interesting that on AML-12 study pts w sAML also had better OS with HiDAC-based induction. In most places, SD-araC is still used, but for sAML or otherwise high risk AML a day 14 marrow would be done and if positive without substantial reduction in blasts – then HiDAC may then be given. Was this common practice on these trials that did not include HiDAC upfront with the initial induction? And if so, is this any different than giving it initially?
Re. We thank the reviewer for this comment. Day-14 bone marrow was not common practice in EORTC trials in which response to induction was generally assessed by day 31 of induction. We have added this in the manuscript (see manuscript lines 282).
- Some of the trials included etoposide or mitoxantrone, or GO, which is a bit confounding. However, this is going to be an issue with any large retrospective study combining trials.
Re. We thank the reviewer for this comment and we agree. We have added this comment in the limitation part of the manuscript (see manuscript lines 255-260).
- Total and in either age groups, <=60 and >60, was there any difference in outcome if pts had sAML from MDS (without prior therapy) vs sAML from prior chemo/rads? I noted there were more fav risk and less adverse in therapy related AML group, which at least in part makes sense biologically as some of the translocations with be CBF etc.
As indicated in Table 2, in younger patients (age <=60 years), group B has similar prognosis as compared to those in group A, once one adjusts by different prognostic features: the HR (group B vs group A) was 0.84 (P=0.20) in model 1, and 0.82 (P=0.13) in model 2.
In older patients, the outcome is dismal in both groups, A and B. The only independent prognostic factors were cytogenetic features, initial performance status, WBC and year of diagnosis, as indicated in the body text (lines 202-203).
- The fact that females has much better OS is very interesting and could be further explored. Also, not unexpected but notable that higher WBC associated with worse survival in sAML, as this usually more associated with de novo disease.
As indicated in Table 2, the prognosis of females was better than of males in younger patients (age <= 60 years). The estimated hazard ratio for OS of females vs males was 0.72 (P=0.01) in the model 1, and 0.68 (P=0.003) in the model 2. These findings are in line with the Swedish registry study of AML patients (https://onlinelibrary.wiley.com/doi/epdf/10.1002/ajh.23908). The estimated OS hazard ratio of males vs females was 1.31, corresponding to HR (females vs males) of 0.76. In contrast, in older patients (age > 60 years), the outcome is similarly dismal in females and males. We agree that the association of higher WBC with poor outcome in sAML patients is also of interest and we have emphasized this finding in the conclusion part of the manuscript. See manuscript lines 3192.
- The 5-yr OS rates post allo and post AUTO HSCT seem quite good. Were they just all younger pts? Any more to be made of this, esp the auto-transplant data?
Re. Indeed the outcome of these patients was quite good. These results are of course limited by the relatively small number of patients in each group and by selection bias since this approach was proposed only in patients who reach CR1, generally younger patients and who remained fit after induction / consolidation chemotherapy.
- Was there any difference in outcome among the t-AML pts based on the prior malignancy type? Or for prior heme vs. solid tumor malignancy?
There was a non-significant in OS between those with prior hemato vs those with solid tumor malignancies.As indicated in the manuscript (and below), the types of prior malignancies were a quite heterogeneous. For those with prior solid tumors, there was a relatively similar OS (HR ranging from 0.71 to 1.25 for the OS of the most common prior malignancies (N=15-17) compared to the OS of patients with a prior breast cancer). Of course, we have to keep in mind that the subgroups are small, so the estimates of the relative difference is wide. For this reason we refrained ourselves to present the results of these subgroups analyses in the manuscript.
“Among other malignancies, breast (n=55), genitourinary (n=17), gynecologic (n=16), head and neck (n=15) and skin (n=10) cancers were the most frequent solid tumors (n=131), while Hodgkin’s lymphoma (n=26) and mature B-cell neoplasm (n=15) were the most frequent other hematologic malignancies (n=45). “
- Finally, the 5-yr OS rate was much worse in over 60, not unsurprising. How many of these pts were transplanted and does this in part explain this? Possibly this reflects that fewer older pts were transplanted in the eras that these trials were done. Biologically the disease “should” be the same, so is this just driven by worse PS/organ function and less application of allo-BMT, or is there some other biologic/epigenetic difference between over 60 and under 60, even if they both have sAML?
Among elderly patients (age > 60 years), who reached CR (N=224), there were only 10 patients who received an autoSCT in CR. Among them, all but two patients died.
AlloSCT was not proposed in our studies in AML elderly patients. Only in the last 10 to 15 years, AlloSCT using a reduced conditioning stated to be used more and more in transplant center, and implemented in several elderly AML clinical trials. This was the case in the EORTC 1301 trial.
Finally, we would like to thank very much the reviewer for his constructive criticisms that helped us improving our manuscript.